# Graphene-Based Films: Fabrication, Interfacial Modification, and Applications

**DOI:** 10.3390/nano11102539

**Published:** 2021-09-28

**Authors:** Sihua Guo, Jin Chen, Yong Zhang, Johan Liu

**Affiliations:** 1SMIT Center, School of Mechatronics Engineering and Automation, Shanghai University, 20 Chengzhong Rd., Shanghai 201800, China; shguo@shu.edu.cn (S.G.); yongz@shu.edu.cn (Y.Z.); 2Electronics Materials and Systems Laboratory, Department of Microtechnology and Nanoscience, Chalmers University of Technology, Kemivägen 9, SE-41296 Gothenburg, Sweden; jin.chen@sht-tek.com; 3SHT Smart High-Tech AB, Kemivägen 6, SE-41258 Gothenburg, Sweden

**Keywords:** graphene-based film, interface modification approach, preparation strategy, thermal and mechanical property

## Abstract

Graphene-based film attracts tremendous interest in many potential applications due to its excellent thermal, electrical, and mechanical properties. This review focused on a critical analysis of fabrication, processing methodology, the interfacial modification approach, and the applications of this novel and new class material. Strong attention was paid to the preparation strategy and interfacial modification approach to improve its mechanical and thermal properties. The overview also discussed the challenges and opportunities regarding its industrial production and the current status of the commercialization. This review showed that blade coating technology is an effective method for industrial mass-produced graphene film with controllable thickness. The synergistic effect of different interface interactions can effectively improve the mechanical properties of graphene-based film. At present, the application of graphene-based film on mobile phones has become an interesting example of the use of graphene. Looking for more application cases is of great significance for the development of graphene-based technology.

## 1. Introduction

Since the successful separation of a few sheets of graphene from graphite using scotch tape in 2004, graphene has gradually entered the life of human beings [1]. Despite of the fact that more than a decade has passed, it is still a hot research topic owing to its extraordinary thermal, electrical, mechanical, and optical properties [2,3,4,5]. Graphene has the potential to be used in many fields such as electronics [6], optoelectronics [7], and electrochemical batteries and composites among many other applications [8]. Graphene-based film has attracted significant attention in practical applications. It has been identified and used for volume application in electronics/battery heat spreading in mobile phones. Thus, a huge interest has emerged in developing this technology on a commercial basis.

However, despite the remarkable performance of single-layer graphene, it is difficult to extend this performance to a multilayer film structure. One of the key weaknesses of the graphene film is the interfacial strength between the graphene layers, causing easy delamination.

Graphene oxide, well known as a precursor for preparing graphene, has abundant oxygen-containing functional groups [9]. Specifically, it has hydroxyl and carbonyl groups on the basal plane and carboxyl groups on the sheet edge [10]. These functional groups can be used as active sites for GO chemical modification and functionalization [11]. Various materials were introduced to GO by covalent or non-covalent bonding to improve its performance. Therefore, many methods were proposed to prepare functionalized graphene-based film [12,13,14].

In this review, we presented and discussed the recent advances in fabrication, processing methodology, the interfacial modification approach, and state of the art applications of graphene-based film. The key focus was on methods of fabrication of the graphene-based film, the interfacial modification concept and strategies to improve its mechanical properties, as well as application challenge and status of graphene-based film’s applications in energy storage and environmental and smart-device applications.

## 2. Preparation Techniques of Graphene-Based Film

The main methods to prepare graphene-based film include vacuum filtration, evaporation-induced self-assembly, rod coating, spin coating, and electro-induced preparation.

### 2.1. Vacuum Filtration

Since Dikin et al. [15] used membrane filters to prepare an ultra-strong graphene oxide paper for the first time in 2007, and vacuum filtration has widely been used in academic research. Many researchers prepared cellulose nanofiber/graphene-based composite film via vacuum filtration in recent years [16,17,18,19,20]. In the electrical shielding application, where electrical insulating materials are required, the excellent electrical conductivity of graphene has become an inevitable obstacle to its application [21]. Therefore, it is important to prepare graphene-based film with insulating properties. Guo et al. [22] fabricated a f-Al_2_O_3_@RGO/nanofibrillated cellulose (f-Al_2_O_3_@RGO/NFC) composite film via a vacuum filtration process. The preparation process is shown in Figure 1a. The introduction of Al_2_O_3_ can not only make the composite film have electrical insulation properties but also increases the thermal transfer path of the composite film. Figure 1b shows the macroscopic photograph and cross-sectional SEM views of the composite film, revealing that the composite film has an ordered and compact structure. The thermal conductivity of cellulose nanofiber/graphene-based composite film is generally low; it is difficult to meet the heat dissipation requirement of modern electronics devices. To achieve high thermal conductivity, carbonization through a high-temperature annealing approach was reported [23]. It was reported that the carbonized polydopamine nanoparticle reinforced graphene films exhibited high thermal conductivity of 1584 W m^−1^ K^−1^ through the carbonized process, as shown in Figure 1c [24].

Vacuum filtration, by which different thickness films can be obtained through different concentration optimizations, is easy to operate. However, it still has many limitations. In the filtration process, the filtration efficiency gradually decreases with the increase of the thickness of the film. In addition, the size of the film is usually limited by the size of the filter. This method seems only to be suitable for the laboratory scale, but not for industrial-scale production.

### 2.2. Evaporation-Induced Self-Assembly

This technology was used to prepare flexible free-standing macroscopic graphene-based films [25,26,27,28,29]. The general process is to pour the slurry into a polytetrafluoroethylene mold or other substrate and then evaporate the solvent at room temperature or a certain temperature. After that, the film is peeled off from the substrate. Capillary-forced-assisted self-assembly is a method for the preparation of reduced graphene oxide film with an unidirectional arrangement [30,31]. The film is initiated at the contact line of the air-liquid-solid interface. Compared with traditional vacuum filtration, scalable large-area highly ordered films are produced by this technique. Wang et al. [32] fabricated graphene films (GFs) with different thicknesses via an evaporation self-assembly process on the aluminum substrate by adjusting the concentration and volume of GO suspension. They found that GFs displayed a favorable thermal conductivity of 3200 W m^−1^ K^−1^ with a smallest thickness of 0.8 μm and which outperformed the Polylitic Graphite Sheets (PGS) by 60%. Besides, GFs show high mechanical tensile strength and excellent flexibility (Figure 2).

The evaporation-induced self-assembly technique has great potential for the industrial-scale production of graphene-based film. This technique is easy to operate without any special equipment. Different-sized films can be obtained by adjusting the size of the substrate. The top surfaces of the film prepared by this method exhibited less roughness than the film fabricated by vacuum filtration [31]. However, it is noteworthy that bubbles may be generated in the evaporation process, which could damage the structure of the film.

### 2.3. Blade Coating

Blade coating is widely used in the laboratory and industrial production [33,34,35,36,37]. In a typical laboratory process, the slurry is transferred to the substrate by moving the blade on the substrate. In industry, however, the slurry is transferred to the fabric through a blade by moving the fabric substrate, and then a freestanding film roll is obtained through multi-stage heat treatment. The thickness of the film can be adjusted by controlling the scraping blade interval. For example, Chen et al. [38] reported a quasi-industrial production of ultra-thick and dense laminated-structures graphene films (GFs) with large sizes and different thicknesses, which were prepared by the blade coating process (Figure 3a,b). In practical industrial applications, two pieces of 220 μm GF and PGF were processed in a standard template of mobile phone and placed on a constant heat source, revealing the GF has faster thermal transfer performance, as shown in Figure 3c. Compared with PGF, the surface of GF still had the original structure after the bending test, which does not hinder the heat diffusion (Figure 3d).

Generally, blade coating is widely used in the current industrial production of graphene film and it has many advantages. For example, this technology can save time, while vacuum filtration is time-consuming. Secondly, it can realize industrial mass production. Additionally, it can precisely control the thickness and size of the film. However, this technology has very strict requirements for the concentration and viscosity of the slurry. If the viscosity of the slurry is too large or too small, the blade coating will be uneven, resulting in an uneven thickness and density of the graphene films.

### 2.4. Spin Coating

The spin coating method is also a common method for preparing graphene-based film [39,40,41,42,43,44,45]. In a typical process, dropping a certain solution on the substrate, continuous thin film is formed under the action of centrifugal force. Spin coating is one of the popular methods of the Layer-by-Layer assembly strategy, which can form a highly ordered multilayer internal structure in a short deposition time [46]. Due to the weak van der Waals interaction among GO nanosheets, the stacked GO layers can be easily separated during the thermal reduction process and numerous voids can be doped, resulting in a significant reduction of cross-plane thermal conductivity of the films [47,48]. However, spin-assist LBL (SA-LBL) utilizes alternating electrostatic deposition between complementary charged materials to construct strong internal bonding multilayers. For example, Hong et al. [49] fabricated an rGO/alumina film via the SA-LBL process, resulting in enhanced in-plane and out-plane thermal conductivity of the composite film. Besides, the SA-LBL process can produce a hydrogen bonding interaction. For example, Song et al. [50] used SA-LBL process to prepare a silicone rubber/graphene film, as shown in Figure 4. Hydrogen interaction and van der Waals forces synergistically increase the bridging between SR and GO to improve the thermal conductivity of the composite film.

Spin coating is a simple, cheap, and versatile technique for preparing ordered multilayered film. However, it is worth noting that the efficiency of spin coating is lower than that of the blade coating. It can only achieve a very thin film at a time. To achieve the same thickness of the film, the spin coating needs to be repeated several times, which is unsuitable for efficient industrial production.

### 2.5. Electro-Induced Preparation

In recent years, electro-deposition has also been used for preparing graphene-based film [51,52,53,54,55,56,57,58]. In the electro-deposition process, the electric field between the two electrodes induces the migration of charged particles and subsequent deposition in a stable suspension [59]. Different foils were fabricated by adjusting electro-deposition parameters such as current density, electro-deposition time, electrolyte concentration, and so on [60].

Because of the existence of oxygen-containing functional groups on GO, GO has a negative charge and good hydrophilicity in water, so GO is generally used as the precursor during the electrodeposition process [51]. The results show that the presence of Cu atoms is beneficial to the reduction of GO [61,62]. Electrolysis of a Cu ion on the Cu anode can remove part of the oxygen-containing functional groups on GO, and the Cu-O-C bonding can be formed between the residual oxygen on RGO and Cu, resulting in the enhancement of the interfacial bonding strength of the Cu-RGO composites. For example, Li et al. [61] prepared the Cu-RGO film by DC voltage at 30 V with different electro-deposition times in GO suspension, as shown in Figure 5. The results indicated that Cu-RGO film exhibits excellent heat-transfer properties and flexibility. Meanwhile, the thermal conductivity of Cu-RGO film increased with the increase of deposition time, which was attributed to the formation of thermal conduction paths in the Cu-RGO film.

Electro-induced deposition is a universal technology that can be applied in any stable suspension. This technology has many advantages, such as good sample uniformity, thickness control, a simple operation process, ease of use, and high cost-effectiveness. The process of preparing graphene-based film by electro-deposition technology also has some drawbacks. It puts forward a higher requirement for the precursor solution. Otherwise, the phenomenon of agglomeration or reduction of deposition efficiency will occur during this process.

## 3. Interfacial Modification of Graphene-Based Film to Improve Its Mechanical Property

It turns out that functionalization is key to make a strong graphene film, especially when several layers are stacked together. The methods mainly include π-π interaction, hydrogen bonding interaction, ionic bonding interaction, covalent interaction, and synergistic interaction.

### 3.1. π-π Interactions

Compared with other interactions, the π-π interaction will not be weakened by the reduction of active sites on RGO. On the contrary, the restoration of the sp^2^-conjugated network and the large graphite domains on graphene can help to form a stronger π-π interaction, significantly improving the interface strength of the graphene film [63]. At the same time, π-π bonding retains the π-conjugated structure of graphene to facilitate charge transfer, improving the conductivity of graphene [64]. Molecules with rich π orbital functional groups such as phenyl and pyrene groups can be used as crosslinking agents to bind the adjacent graphene sheets [65,66]. For example, Ni et al. [67] synthesized AP-DSS molecules with pyrene groups at both ends. AP-DSS was used as a cross-link to improve interface interaction of adjacent rGO nanosheets through π-π interaction, as shown in Figure 6a. During the loading, AP-DSS molecules stretched along with the slip between the graphene nanosheets, and a significant amount of energy was absorbed. With further loading, the π-π cross-link structure between the pyrene group and graphene sheets was destroyed, and a large amount of energy was further absorbed. Finally, the graphene nanosheets were pulled out, resulting in edge-curing of graphene nanosheets (Figure 6b,c). The maximum tensile strength of rGO-AP-DSS composites film can reach 538.8 ± 31.6 MPa, which is about 4.1 times higher than that of pure rGO film, as shown as Figure 6d. Meanwhile, the electrical conductivity of the composite film was also significantly improved. However, compared with small molecules, the slippage distance of graphene nanosheets increased sharply when long-chain molecules were used as a crosslinking agent, resulting in a significant improvement in mechanical properties. Wan et al. [68] used long-chain bis (1-pyrene methyl) docosa-10,12-diynedioate (BPDD, C_16_H_9_CH_2_OOC(CH_2_)_8_C≡C-C≡C (CH_2_)_8_COOCH_2_C_16_H_9_) monomers to π-π bond adjacent graphene nanosheets, which exhibited an ultrahigh tension strength of 1054 MPa.

### 3.2. Hydrogen Bonding

Spider silk has excellent mechanical properties and is superior to most other fibers due to the multi-scale hierarchical structure formed by multiple hydrogen bonds between its proteins [69]. Although the bond energy of hydrogen bonds is weaker than that of the covalent bonds, multiple hydrogen bonds can also significantly improve the performance of materials [70]. Inspired by nature, polymers containing multiple hydrogen bonds such as cellulose, sodium alginate, calcium alginate, poly (vinyl alcohol), and chitosan, etc. are used to hydrogen bond adjacent graphene nanosheets to improve their mechanical properties [71,72]. For example, RGO/CA (calcium alginate) film was demonstrated by Jia et al. [73] through a vacuum-assisted assembly. The cross-section surface morphology of composite film showed an oriented structure. The tensile strength of 118 MPa and toughness of 4.6 MJ/m^3^ were attributed to the synergistic interaction of the hydrogen interaction between CA and rGO and ionic bonding between calcium ions and rGO nanosheets. Moreover, the film showed a high EMI shielding reliability. Furthermore, Duan et al. [74] reported a reduced graphene oxide-nanofibrillar cellulose-10,12-pentacosadiyn-1-ol (rGO-NFC-PCDO) ternary composite film with layered structure (Figure 7a). Because the abundant hydroxyl groups on NFC react with the oxygen-containing group on GO to form hydrogen bonding, introduced PCDO can covalently cross-link GO through esterification. RGO-NFC-PCDO ternary composite film has a tensile strength of 314.6 ± 11.7 MPa and a toughness of 9.8 ± 1.0 MJ/m^3^, which are higher than that of other GO-based films, as shown in Figure 7b. Under stress, the hydrogen bond between rGO and NFC is first broken, and the stress is uniformly dispersed in rGO and NFC. At the same time, PCDO molecules consume more energy in the stretching process. During further stretching, the NFC chain is pulled out. Then, the covalent bonding between PCDO and rGO is destroyed (Figure 7c).

### 3.3. Ionic Bonding

Various oxygen-containing functional groups such as hydroxyl, epoxide, and carboxyl groups are distributed on the surface and edge of GO sheets. Due to the ionization of hydroxyl and carboxyl groups, GO has a strong negative charge when dispersed in water [75]. GO may interact with positively charged metal ions [76]. In nature, the report found that the presence of small amounts of metal elements, (such as Zn, Mn, Ca, or Cu) in insects or other organisms can significantly increase their mechanical strength [77]. Inspired by this phenomenon, Mg^2+^, Cu^2+^, Ca^2+^, Zn^2+^ and other metal ions were incorporated into the internal structure of GO film to form ionic bonds with oxygen-containing functional groups on GO, which significantly improved the mechanical properties and stability of the graphene-based film. For example, Ruoff et al. [78] modified a graphene oxide paper with a small amount of Mg^2+^ or Ca^2+^. Because metal ions were tightly bonded to the carboxyl group on edge of GO sheets, a small amount of Mg^2+^ or Ca^2+^ content (less than 1 wt%) can significantly improve the mechanical stiffness (10–200%) and fracture strength (50%) of GO paper, as shown in Figure 8a. Furthermore, Xing et al. [76] introduced the Ca^2+^ into graphene oxide/Sodium alginate (GO/SA) film via a facile and environmentally friendly method. Because the adjacent graphene oxide sheets were bridged by the SA and Ca^2+^ through hydrogen bonding and ionic bonding interaction, Ca-GO/SA film with the tensile strength of 97 ± 4 MPa was stronger than GO/SA film with a tensile strength of 80 ± 7 MPa (Figure 8b,c). There are many reports about the use of Ca^2+^, but some researchers also introduced Cu^2+^ into the graphene-based film. For example, Wang et al. [79] constructed an ultra-robust and high-toughness GO paper via the synergistic reinforcement of CNTs and Cu^2+^. Compared with the GO papers, the tensile strength, elastic modulus, and toughness of the Cu-CNTs/GO paper was increased by 409.7%, 81.5%, and 188.2%, respectively. Cu-CNTs/GO composite paper has π-π interactions between CNTs and GO and ionic bonding between Cu^2+^ and the carboxyl groups on GO sheets or CNTs, as shown in Figure 8d. During the tensile process, the synergistic effect of π-π interactions and ionic bonding interactions enhance the interaction among GO sheets and promote the uniform stress transfer in the composite paper. During the fractures, due to the synergistic inhibition of CNTs and Cu^2+^, the slippage of GO sheets is restrained compared with the GO paper (Figure 8e).

### 3.4. Covalent Bonding

It was found that borate was used to covalently cross-link RG-II in higher plants to improve the mechanical strength of cell network of the plant [80]. An et al. [81] used borate ions to covalently cross-link adjacent graphene oxide sheets and combine low-temperature annealing. Borate ortho-ester bonds were formed by the reaction of borate ions with hydroxyl groups on GO sheets, and more covalent bonds were formed during the thermal annealing process, which improved the mechanical properties more than that of the unmodified film (Figure 9a). In addition, some researchers have used glutaraldehyde (GA) as a crosslinking agent to introduce into the graphene-based film, forming a covalent crosslink among adjacent graphene nanosheets. For example, Gao et al. [82] introduced GA and H_2_O into GO sheets to strengthen the interlayer adhesion. Aldehyde groups on GA reacted with hydroxyl groups on GO sheets through intermolecular acetalization, enhancing covalently crosslinked adjacent GO sheets. Water molecules can not only weaken the adhesion between layers, but also act as a lubricant to promote the GO sheets slippage during loading. Therefore, the combined use of GA and H_2_O can make the properties of GA-H_2_O-treated GO superior to other composite films.

However, GA is toxic. Therefore, it is necessary to develop natural green products as crosslinking agents to improve the mechanical properties of the graphene-based film. Chitosan (CS) is the product of removing part of acetyl group from chitin, which has many advantages such as biodegradability, non-toxicity, and having many amino groups and hydroxyl groups [83]. For example, Wan et al. [84] demonstrated a strong and tough graphene-chitosan (rGO-CS) film. The tensile strength, toughness, and electrical conductivity of the rGO-CS film reached 526.7 MPa, 17.7 MJ/m^3^ and 155 S/cm, respectively, owing to the synergistic interactions of hydrogen and covalent bonding. Thus, it can absorb more energy to promote stress transfer and achieve a high tensile strength and toughness of composite film (Figure 9b). In addition, dopamine is a kind of material containing amino and catechol functional groups, and it can self-polymerize to long-chain polydopamine (PDA) under alkaline conditions. PDA has a strong bonding ability to any substance, and its abundant amino groups and catechol groups can form stable covalent bonds with oxygen-containing functional groups [85]. Cui et al. [86] developed a strongly integrated and high-toughness graphene oxide film via dopamine covalent cross-link. The maximum tensile strength and toughness of rGO-PDA film were 204.9 ± 17.0 MPa and 4.0 ± 0.9 MJ/m^3^ respectively, which is higher than that of the pure GO film. Figure 9c exhibits the structural model for dopamine cross-link graphene oxide film. Adjacent GO sheets were crosslinked by PDA, provided there was enough space for GO sheets slippage, and absorbed more energy during loading.

**Figure 9 nanomaterials-11-02539-f009:**
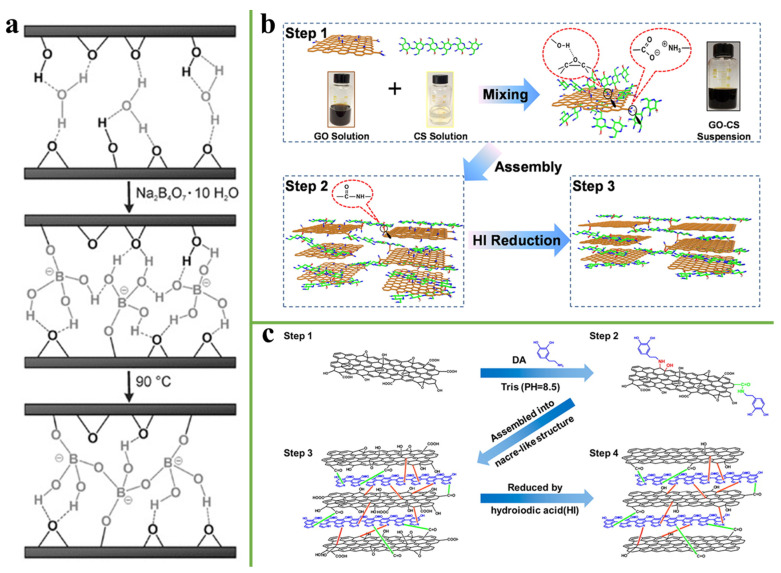
(**a**) Schematic illustration of the formation of the borate-crosslinked network across two adjacent graphene oxide nanosheets. Reprinted with permission from ref. [81] Copyright 2011, Wiley. (**b**) Illustration of the manufacturing process of rGO-CS artificial nacres. Reprinted with permission from ref. [84] Copyright 2015, American Chemical Society. (**c**) Structural model for dopamine cross-linked graphene oxide film. Reprinted with permission from ref. [86] Copyright 2014, American Chemical Society.

### 3.5. Synergistic Interaction

Nature materials such as nacre, bones, and silk, etc. are composed of hard and soft phases arranged in a complex hierarchical structure, showing unique strength and toughness. Therefore, the layered design in nature is an effective way to improve the mechanical properties of materials [87]. Inspired by this, the combination of different interfacial interactions can synergistically enhance the mechanical properties of the graphene-based film. For example, Li et al. [88] reported a strong and tough rGO/NFC/PDA ternary composite film. Because of the synergistic interaction of hydrogen bonding, ionic bonding, and covalent bonding, RGO/NFC/PDA ternary composite film has a tensile strength of 528 MPa and toughness of 7.3 MJ/m^3^, which are superior to other GO-based composite films.

Super-tough graphene oxide/sulfonated styrene ethylene/butylene-styrene (GO-SSEBS) film was demonstrated by Song et al., as shown in Figure 10a [89]. The tensile strength and toughness of GO-S-10 reached 158 ± 6.0 MPa and 15.3 ± 1.5 MJ/m^3^, which are 76% and 900% higher than pure GO, owing to the π-π interaction formed between GO and poly-styrene (PS) in SSEBS and hydrogen bonding formed between oxygen-containing groups on GO with sulfonic acid groups on SSEBS. GO-S-10 has two failure stages in stress-strain curves, including plastic deformation and a hardening stage compared with GO. At the initial loading, the GO-S-10 first undergoes plastic deformation. The EB soft chain in SSEBS begins to extend from the randomly coiled conformation between GO sheets, resulting in large energy dissipation. With the continuous stretching, the EB segment extends further. Then, the hardening stage occurs and the EB chain begins to fracture (Figure 10b). Furthermore, Zhou et al. [90] used Mxene sheets to functionalize graphene oxide via Ti-O-C covalent bonding to obtain MrGO and 1-aminopyrene-disuccinimidyl (AD), which were then used to crosslink MrGO through π-π interaction to obtain MrGO-AD composite film (Figure 10c). The MrGO-AD composite film has an ultrahigh tensile strength of 699.1 ± 30.6 MPa, a failure strain of 12.0 ± 0.7%, and a toughness of 42.7 ± 3.4 M/J m^3^, superior to that of other GO-based films.

γ-poly(glutamic acid) acid (PGA) is a naturally occurring polymer that can be produced by bacteria [91]. It contains many -COOH and -NH_2_ functional groups. Liang et al. [92] fabricated a layered GO/PGA/Ca^2+^ composite film via synergistic interaction of hydrogen bonding between GO and PGA and ionic bonding between GO and Ca^2+^, as shown in Figure 11a. The composite film exhibited a tensile strength of 150 ± 51.9 MPa and an outstanding Young’s modulus of 21.4 ± 8.7 GPa, representing an enhancement of 120% and 70% compared with GO film, respectively. Liang et al. and many previous reports immersed the prepared graphene-based film into the metal solution to synergistically improve the interfacial interaction of film. However, the ionic bond is usually chelated by GO and exists in the intermediate layer of GO nanosheets. The relatively weak ionic bond inevitably limits the load transfer of the GO nanosheets under stress [93]. In the jaw of Glycera, a small amount of copper ions are present in proteins, chelated by the imidazole group of histidine [94]. The formed metal-ligand coordination bond is half the strength of the covalent bond and it is highly cross-linked into the protein, which is beneficial to load transfer [95]. Inspired by this, Cheng et al. [93] introduced Cu^2+^ into CS to form a metal-ligand coordination bond, and mixed it with GO solution to prepare a rGO/CS/Cu^2+^ composite film, as shown in Figure 11b. The tensile strength and electrical conductivity of the rGO/CS/Cu^2+^ composite film reached 868.6 MPa and 234.8 ± 14.4 S cm^−1^, respectively. The binding force between Cr^3+^ and oxygen-containing functional groups is stronger than other alkaline earth-metal ions [96]. Wang et al. [97] described a new continuous bridging strategy in which GO nanosheets were first bridged by Cr^3+^ and rGO was then π-π bonded through PSE-AP to obtain a SBG composite film (Figure 11c). The SBG film exhibited an excellent tensile strength of 821.2 MPa, a toughness of 20.2 MJ m^−3^, and an electrical conductivity of 415.8 S cm^−1^, which are 4.0, 7.5, and 1.9 times higher than that of rGO, respectively.

## 4. Applications

With the continuous increase of energy consumption, the development of renewable energy has become one of the most important topics. The energy market urgently needs electrochemical energy storage devices with efficient energy storage and conversion capabilities [98]. The growing demand of flexible supercapacitors has aroused great interest. Because of its high electrical conductivity, large specific surface area, and especially its strong mechanical properties, graphene-based film is considered a promising electrode material for supercapacitors [99]. Song et al. [100] fabricated a graphene-based film by introducing an activated-carbonized cotton fiber (ACC) to regulate the chemical composition, surface area, and pore size distribution. They found that ACC-rGO film exhibited an enhanced energy storage capability (capacitance of 310 F g^−1^ and 150 F g^−1^ at 0.1 A g^−1^ and 10 A g^−1^, respectively), an excellent power density of 156.5 mW cm^−2^, and an energy density of 240 μWh cm^−2^ (Figure 12a). Li et al. [101] developed a flexible all-solid-state supercapacitor of graphene/MoS_2_ film with a volumetric capacitance of 19.44 F cm^−3^. Moreover, it could maintain 87% of its original capacitance after 300 stretch cycles, which exhibited excellent stretchability and stability compared with most other supercapacitors. With the rapid development of wearable electronic devices, the demand for flexible lithium-ion batteries is increasing. The performance of lithium-ion batteries largely depends on electrode materials. Graphene is also considered as a suitable matrix for the formation of flexible electrodes. Zhou et al. [102] demonstrated a lamellar graphene/nanocellulose/silicon (GN/NC/Si) film through covalent crosslinking of glutaraldehyde (Figure 12b). When used as an anode, GN/NC/Si film has a high reversible capacity of 1251 mAhg^−1^ at 100 mAg^−1^ after 100 cycles and an excellent rate capability. Moreover, the film represents robust mechanical strength and good flexibility. Furthermore, a free-standing flexibility Li_4_Ti_5_O_12_-rGO (LTO-rGO) film was fabricated by Zhu et al. [103]. The LTO-rGO electrode had good electron/ion conductivity and mechanical properties, an enlarged electrode/electrolyte contact area, and an excellent specific capacity of 135.4 mAhg^−1^ at 40 C.

The greenhouse gases produced in fossil fuels have caused many environmental problems such as abnormal climates, rising sea levels, and air pollution, which seriously endanger the survival of human beings. Graphene-based film has a unique layered and porous structure, so it has received extensive attention in environmental applications [104]. Among them, graphene-based film is widely used in the application of gas treatment. A novel ZIF-8@GO film for hydrogen selectivity was reported by Huang et al. [105] (Figure 12c). At 250 °C and 1 bar, the mixture separation factors of H_2_/CO_2_, H_2_/N_2_, H_2_/CH_4_, and H_2_/C_3_H_8_ were 14.9, 90.5, 139.1, and 3816.6, with H_2_ permeances of about 1.3 × 10^−7^ mol·m^−2^·s^−1^·Pa^−1^, which is promising for hydrogen separation and purification by molecular sieving. Furthermore, Zeynali et al. [106] also reported a graphene oxide film for hydrogen separation. For the GO film, the H_2_/CO_2_ and H_2_/N_2_ values were 3.83 and 16.5 (H_2_ permeance equal to 5.9 × 10^−7^ mol·m^−2^·s^−1^·Pa^−1^), respectively, while these values at 473 K and 1 bar pressure gradient were 15.7 and 10.6 (H_2_ permeance equal to 7.6 × 10^−7^ mol·m^−2^·s^−1^·Pa^−1^). In addition, graphene-based film also is widely used in the application of water treatment. Grossman et al. [107] used classical molecular dynamic simulations to explore how multilayer nanoporous graphene (NPG) might serve as a reverse osmosis film in water desalination. Compared with single-layer film, multilayer NPG has a similar desalination performance, and its separation performance can be designed by manipulating different configuration variables (Figure 12d). Graphene-based materials have the characteristics of high specific area, high electron mobility, and low electrical noise. In recent years, it has been widely used in sensing applications [108]. At present, a series of sensors have been developed such as chemical sensors, bio-sensors, and gas sensors, etc. Goldsmith et al. [109] developed a cost-effective portable graphene biosensor for the detection of the Zika virus using highly specific immobilized monoclonal antibodies. The percentage of capacitance change in response to antigen dose was consistent with the clinical level, and the antigen detection concentration in buffer was as low as 450 PM (Figure 12e). Furthermore, Zhang et al. [110] used a laser-induced graphene (LIG) electrochemical sensor for detection of trans-resveratrol (TRA) molecules in red wine and grape skin. The LIG sensor had excellent repeatability, stability, reproducibility, and reliability. Moreover, this sensor showed a good linear response in the concentration range of 0.2 to 50 μmol L^−1^, and the lower limit of detection (LOD) was 0.16 μmol L^−1^. Besides, researchers have developed gas sensors. For example, Seekaew et al. [111] reported a novel graphene-based electroluminescent (EL) gas sensor for CO_2_ detection at room temperature. Compared with other graphene-based gas sensors, a graphene-based EL gas sensor can measure the CO_2_ concentration through the change of EL intensity, and it can be directly applied on a smartphone without additional hardware (Figure 12f).

**Figure 12 nanomaterials-11-02539-f012:**
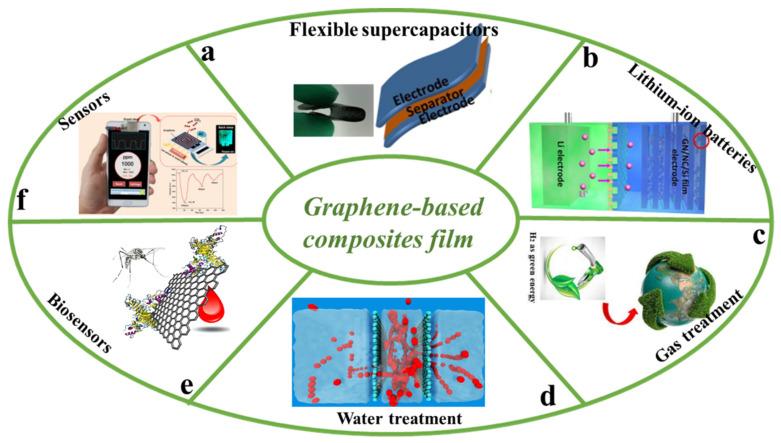
Applications of the graphene-based film. (**a**) Flexible graphene-based composite films for Supercapacitors. Reprinted with permission from ref. [100] Copyright 2017, Elsevier. (**b**) A free-standing sandwich-type GN/NC/Si Laminar Anode for flexible rechargeable Lithium-ion batteries. Reprinted with permission from ref. [102] Copyright 2018, American Chemical Society. (**c**) High-performance graphene oxide (GO) nanocomposite membrane for hydrogen separation. Reprinted with permission from ref. [105] Copyright 2014, American Chemical Society. (**d**) Multilayer nanoporous graphene membranes for water desalination. Reprinted with permission from ref. [107] Copyright 2016, American Chemical Society. (**e**) Novel graphene-based biosensor for early detection of Zika virus infection. Reprinted with permission from ref. [109] Copyright 2018, Elsevier. (**f**) Graphene-based electroluminescent gas sensor for CO_2_ detection at room temperature. Reprinted with permission from ref. [111] Copyright 2019, Elsevier.

## 5. Conclusions and Outlook

This review summarized the research progress of graphene-based film, involving the preparation methods, interface modification approach, and state of the art applications. Various methods such as vacuum filtration and spin coating can be used to prepare graphene-based film. Among them, only blade coating can achieve large-scale industrial production, but the production quantity is still limited. At the same time, the mechanical properties of macro graphene-based film can be enhanced by constructing covalent or non-covalent bonds at the active sites of GO. However, there are also many limitations. Firstly, the interfacial modification method of graphene-based film is extremely complex, costly, and uses a large amount of toxic or harmful solvents, rendering it unsuitable for industrialization. Therefore, it is necessary to develop efficient, cheap, easy-to-operate, and environmentally friendly preparation strategies. Secondly, the current research on the properties of graphene-based film was mainly focused on the mechanical and electrical properties and to a smaller extent on thermal properties. A key challenge is to simultaneously improve the mechanical properties and thermal conductivity of the graphene-based film used in microelectronics devices. The interface modification methods mainly involve some polymers and long or small-chain molecule cross-linking agents, but these cross-linking agents will decompose under high-temperature carbonization and graphitization conditions. At present, the general reduction method of graphene-based film in the literature is HI low-temperature reduction, but high thermal conductivity cannot be obtained. Therefore, the needs still remain to develop graphene-based film with both high mechanical properties and thermal conductivity simultaneously.

With the continuous expansion of the heat dissipation market for radio-base stations, power modules, and terminal electronics products in the 5G era, graphene dissipation film made of graphene oxide as a raw material through the coating, drying, heat treatment, calendaring, and cutting processes becomes a common choice for many manufacturers. After the first application of graphene film in the Huawei Mate20X, the first domestic 5G tablet Huawei MatePadPro5G was equipped with an ultra-thick 3D graphene film with 400 μm. Furthermore, Redmi K40, vivo Z6, and Oppo Reno 3 also used graphene film for heat dissipation. At the same time, iPhone and Samsung also accelerated the application of graphene dissipation technology in terminals. In recent years, companies that produce graphene dissipation film such as Changzhou Fuxi Ltd., Shenrui Moxi Ltd., and Yuntian Morui Ltd., among others, have emerged in China. Their products have entered the supply chain system of mobile manufacturers. However, some of them still suffer from limited production capacity. Therefore, it is important to realize the green, large-scale, stable, and high-quality industrial production of high-performance graphene-based film to meet the needs of the development of large power dissipation needs in 5G, Opto, LED, IGBT, and other applications. The graphene industry has great prospects and growth potential. However, as an emerging industry, the graphene industry needs more development. Only through major-breakthroughs in many technical contexts can it bring disruptive changes. We are looking forward to that day.

## Figures and Tables

**Figure 1 nanomaterials-11-02539-f001:**
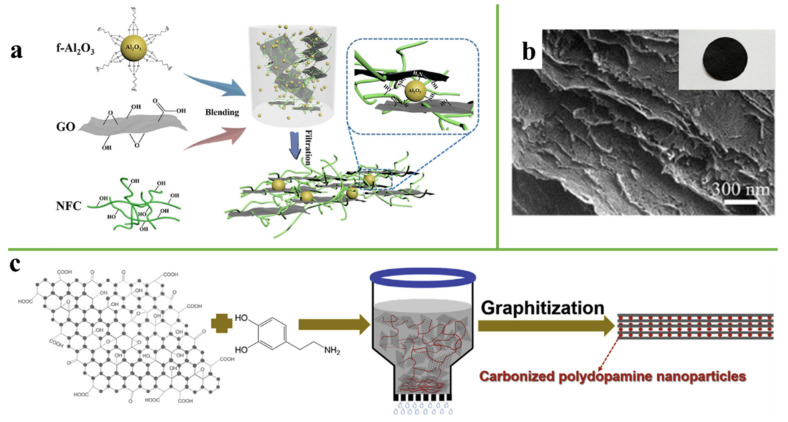
(**a**) Schematic illustration of the fabrication process of a f-Al_2_O_3_@RGO/NFC composite film by vacuum filtration. Reprinted with permission from ref. [22] Copyright 2019, Elsevier. (**b**) Macroscopic photograph and cross-sectional SEM views of composite film. (**c**) Schematic illustration of GF. Reprinted with permission from ref. [24] Copyright 2019, Elsevier.

**Figure 2 nanomaterials-11-02539-f002:**
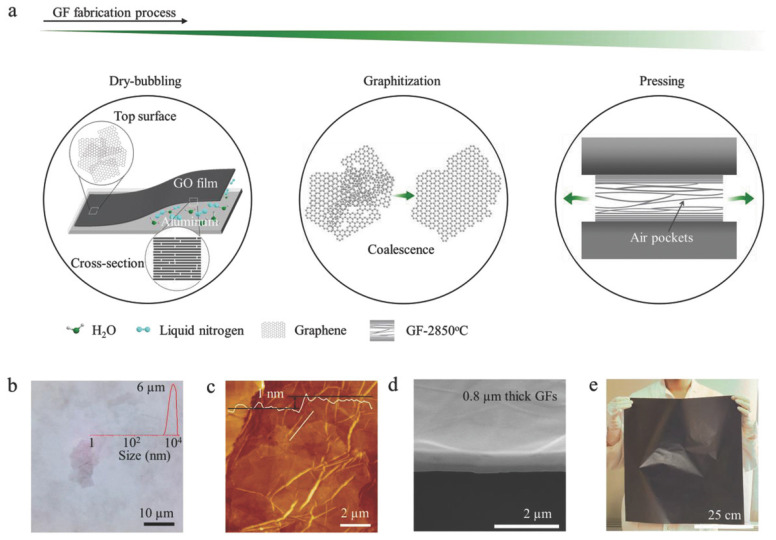
(**a**) Sketch of the fabrication process of GF. (**b**) Optical image of GO flakes with an average size of about 6 µm. (**c**) AFM image of GO flakes with a thickness of less than 1 nm. (**d**) SEM image of cross-section of the fabricated GFs. (**e**) Optical image of the fabricated large-area GFs. Reprinted with permission from ref. [32] Copyright 2018, Wiley.

**Figure 3 nanomaterials-11-02539-f003:**
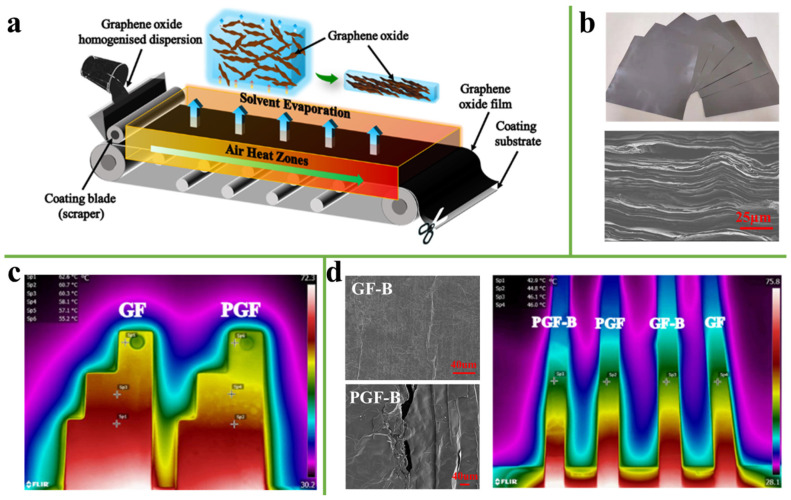
(**a**) The Schematic of the fabrication process of the GO film. (**b**) Optical image of the GFs with different thicknesses and cross-sectional SEM image of pressed GFs-2850. (**c**) Infrared thermal images of 220 μm GF and PGF in a standard template. (**d**) SEM image of the surface morphology change of GF and PGF after the bending test, respectively, and infrared thermal images of GF and PGF before and after the bending test. Reprinted with permission from ref. [38] Copyright 2020, Elsevier.

**Figure 4 nanomaterials-11-02539-f004:**
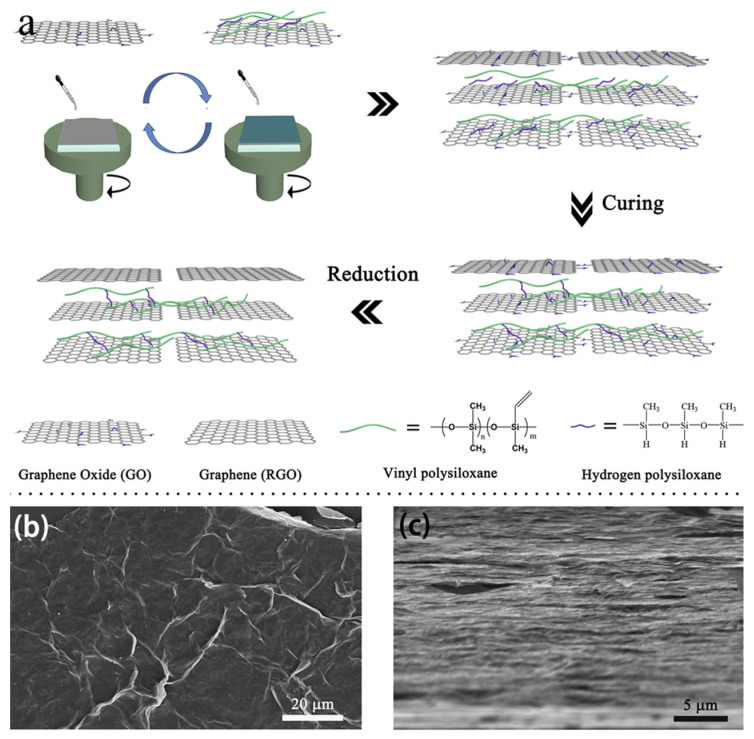
(**a**) Schematic illustration of the proposed fabrication procedure of SR/RGO multilayered films via spin-assisted LBL assembly. (**b**) SEM image of GO on SR. (**c**) Cross-section SEM image of (SR/RGO)_40_ film. Reprinted with permission from ref. [50] Copyright 2018, Elsevier.

**Figure 5 nanomaterials-11-02539-f005:**
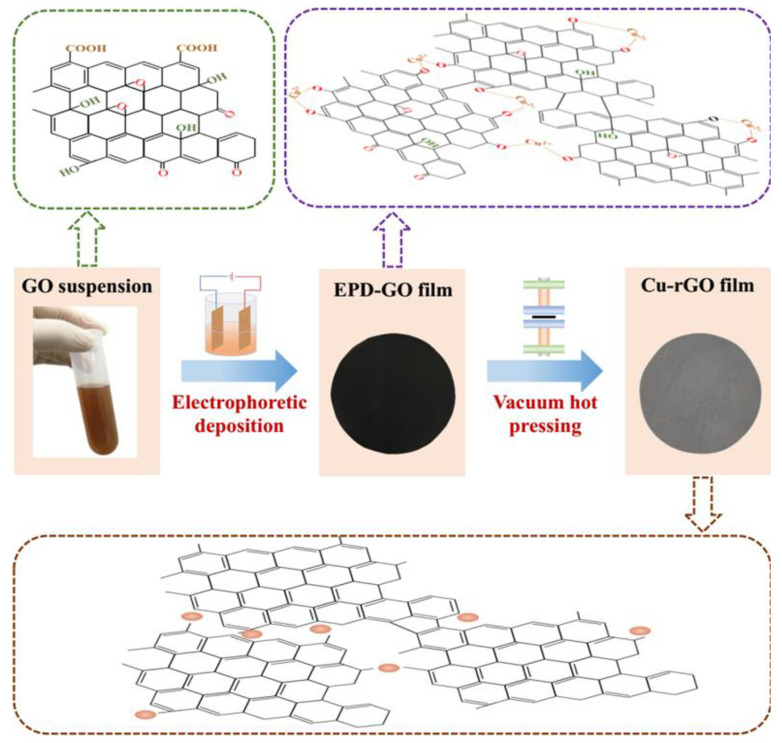
Schematic depicts the preparation process of composite film. Reprinted with permission from ref. [61] Copyright 2020, Elsevier.

**Figure 6 nanomaterials-11-02539-f006:**
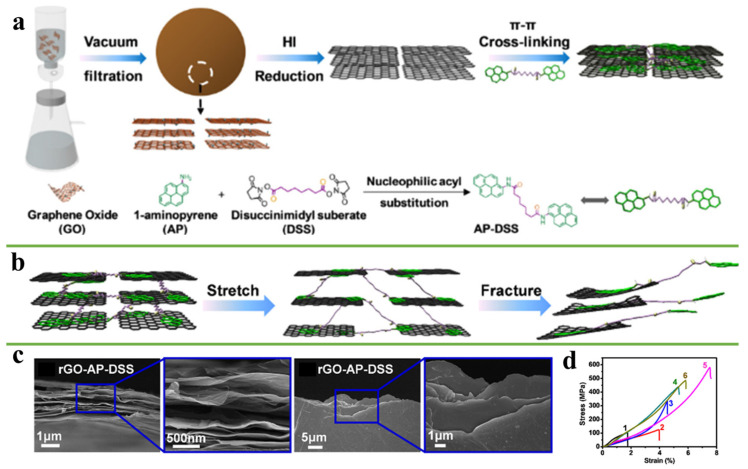
(**a**) Schematic illustration of the fabrication of bioinspired rGO-AP-DSS films. (**b**) Proposed fracture mechanism of rGO-AP-DSS film. (**c**) SEM images of the front- and side-view fracture surfaces of rGO-AP-DSS films after tensile testing. (**d**) Stress-strain curves of GO (curve 1), rGO (curve 2), and rGO-AP-DSS with different contents of AP-DSS designated as rGOAP-DSS-I (curve 3), rGO-AP-DSS-II (curve 4), rGO-AP-DSS-III (curve 5), and rGO-AP-DSS-IV (curve 6). Reprinted with permission from ref. [67] Copyright 2017, American Chemical Society.

**Figure 7 nanomaterials-11-02539-f007:**
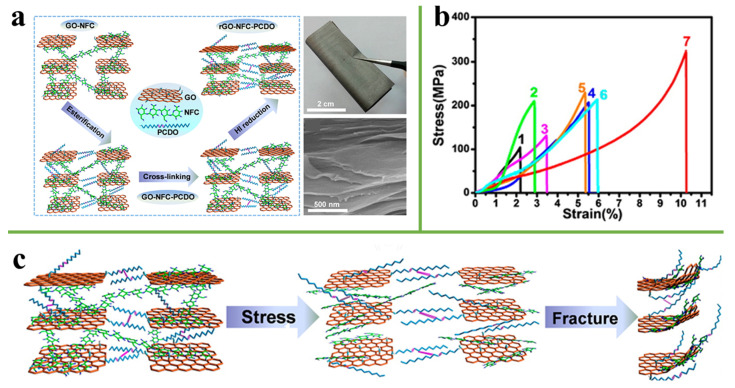
(**a**) Schematic illustration of the fabrication procedure of ternary artificial nacre nanocomposites. (**b**) Stress-strain curves of GO (Curve 1), NFC (Curve 2), rGO (Curve 3), GO-NFC-IV (Curve 4), GO-NFC-PCDO-IV (Curve 5), rGO-NFC-IV (Curve 6), and rGO-NFC-PCDO-IV (Curve 7). (**c**) The fracture mechanism of the rGONFC-PCDO-IV ternary artificial nacre nanocomposite under stress. Reprinted with permission from ref. [74] Copyright 2016, American Chemical Society.

**Figure 8 nanomaterials-11-02539-f008:**
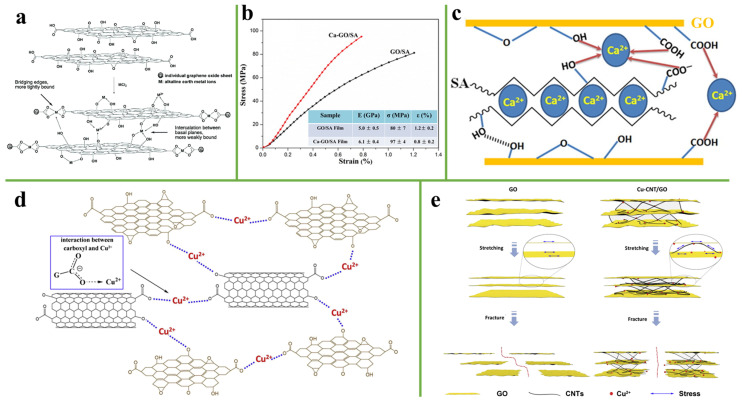
(**a**) Schematic model of the reaction between graphene oxide paper and MCl_2_ in M-modified graphene oxide papers. Reprinted with permission from ref. [78] Copyright 2008, American Chemical Society. (**b**) Stress-strain curves of GO/SA films and Ca-GO/SA films. (**c**) Structural model for metal ion-modified nacre-mimetic film. Reprinted with permission from ref. [76] Copyright 2017, Elsevier. (**d**) Schematic illustration showing the interactions between Cu^2+^ and GO or CNTs. (**e**) Schematic diagram showing the stress transfer mechanisms and fracture behaviors of GO and Cu-CNTs/GO. Reprinted with permission from ref. [79] Copyright 2017, Elsevier.

**Figure 10 nanomaterials-11-02539-f010:**
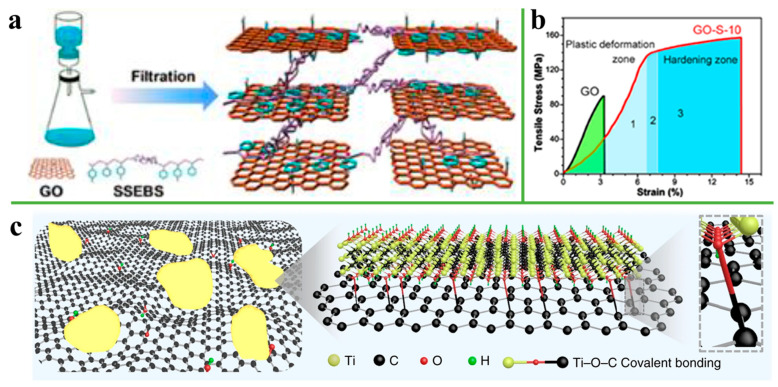
(**a**) Schematic representation of the fabrication process of GO-S artificial nacre via vacuum-assisted filtration. (**b**) Stress-strain curve of the pure GO and GO-S-10 film. Reprinted with permission from ref. [89] Copyright 2017, Elsevier. (**c**) Schematic model of MXene-GO platelets showing the formation of Ti-O-C covalent bonding. Reprinted with permission from ref. [90] Copyright 2020, Springer.

**Figure 11 nanomaterials-11-02539-f011:**
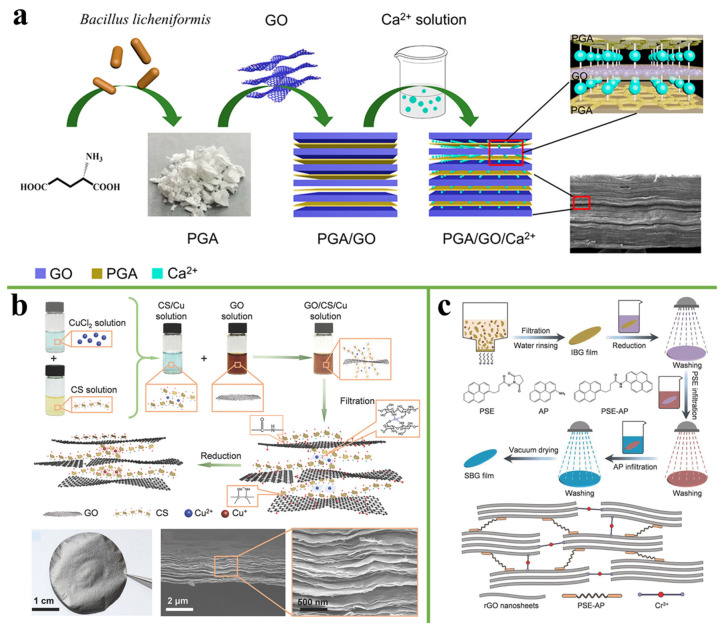
(**a**) Schematic illustration of the preparation process for GO/PGA/Ca^2+^ composite films [92]. Reprinted with permission from ref. [92] Copyright 2020, American Chemical Society. (**b**) The illustration of the fabrication method for rGO-CS-Cu nanocomposites. Reprinted with permission from ref. [93] Copyright 2018, Wiley. (**c**) Preparation and structural characterization of SBG sheets. Reprinted with permission from ref. [97] Copyright 2018, Wiley.

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
