# Peer review of "Graphene-Based Films: Fabrication, Interfacial Modification, and Applications"

_nanomaterials, 2021, doi:10.3390/nano11102539_

Round 1

Reviewer 1 Report

The manuscript considers well the literature on graphene-based films. The involved literature looks well and enough. I propose to improve English deeply. There are lots of minor errors. For example, 'electrical insulting materials'...insulating, or 'with large size and different thickness… with a large size and different thicknesses', etc. Moreover, sections 2.2, 3.2, and 4 were written in different font sizes. From this reviewer's point of view, after these corrections, the manuscript can be accepted for publication.

Author Response

Dear Reviewer

Please find enclosed our group’s manuscript entitled “Graphene-based films: Fabrication, Interfacial Modification, and Applications” by Sihua Guo et al. We thank you spending your time reviewing our manuscript and for this kind comments. We have checked the whole manuscript very carefully and corrected all the mistakes. All changes are highlighted with red color in the revised manuscript. Please see the attachment.

We hope that these modifications make the manuscript suitable for publication in Nanomaterials. We believe that this work will be of broad interest to the readership of the journal.

We thank you in advance for your time and consideration.

Sincerely,

Johan Liu

Reviewer 2 Report

This manuscript is a comprehensive review article under the title “Graphene-based films: Fabrication, Interfacial Modification, and Applications.” It is certainly a good effort to summarize the synthesis methods and potential applications. I will recommend authors to replace the old references with the recent papers published within the last five years to make their review paper up to date with the new trends in recent related research.

Author Response

Dear Reviewer

Please find enclosed our group’s manuscript entitled “Graphene-based films: Fabrication, Interfacial Modification, and Applications” by Sihua Guo et al. We thank you spending your time reviewing our manuscript and for this kind comments. Following your suggestion, we carefully read the literatures and replace many old references with the recent papers published within the last five years in the revised manuscript and highlighted with red color. Since some others references could not be modified, we did not replace them in the manuscript. Please see the attachment.

We hope that these modifications make the manuscript suitable for publication in Nanomaterials. We believe that this work will be of broad interest to the readership of the journal.

We thank you in advance for your time and consideration.

Sincerely,

Johan Liu
